# Antibacterial Activity of Romanian Propolis against *Staphylococcus aureus* Isolated from Dogs with Superficial Pyoderma: In Vitro Test

**DOI:** 10.3390/vetsci9060299

**Published:** 2022-06-16

**Authors:** János Dégi, Viorel Herman, Violeta Igna, Diana Maria Dégi, Anca Hulea, Florin Muselin, Romeo Teodor Cristina

**Affiliations:** Faculty of Veterinary Medicine, Banat’s University of Agricultural Sciences and Veterinary Medicine Timișoara, Calea Aradului 119, 300645 Timișoara, Romania; viorelherman@usab-tm.ro (V.H.); violetaigna@usab-tm.ro (V.I.); diana.maria.degi@gmail.com (D.M.D.); anca.hulea@usab-tm.ro (A.H.); florinmuselin@usab-tm.ro (F.M.); rtcristina@yahoo.com (R.T.C.)

**Keywords:** propolis, staphylococci, dogs, pyoderma, resistance

## Abstract

Staphylococcal infection treatment in dogs is frequently associated with adverse side effects, high costs, prolonged treatment, and resistant strain selection. *Staphylococcus aureus* is the most frequently isolated staphylococci in cases of canine superficial pyoderma. The number of *Staphylococcus* strains to exhibit primary resistance to various drugs in vitro is increasing. Propolis has a diverse chemical composition and well-known therapeutic properties against bacterial infections. The current investigation evaluated in vitro the antimicrobial activity of the commercial allopathic antimicrobials, Romanian propolis ethanolic extracts, against clinical *Staphylococcus aureus* strains isolated from superficial dermatitis clinical samples in dogs and two reference strains: *Staphylococcus aureus* ATCC 25923 and *Staphylococcus aureus* ATCC 43300, as the MSSA and MRSA positive controls, respectively, in western Romania. We used the microdilution broth technique to evaluate the susceptibility profile of the bacteria. The minimum inhibitory concentration (MIC) of the Romanian propolis ethanolic extract ranged from 6 to 10 μg/mL for all isolates, determined by the broth microdilution method. The MICs of ethanolic Romanian propolis extracts had a pronounced antibacterial activity. These results indicate that propolis can potentially be used and recommended for in vivo experiments as a promising therapeutic agent against *Staphylococcus aureus* infections in superficial dermatitis of dogs.

## 1. Introduction

Due to the scarcity of novel and effective therapeutic agents, the pharmaceutical industry has discovered a new source of therapeutic compounds in natural products and herbal medicine, to address current health problems in humans and animals [1,2]. These agents are up-and-coming, because they have fewer side effects and are less expensive than synthetic compounds [2]. This means that these treatments will be more widely available, especially for people in less developed countries who cannot afford expensive treatments. Reduced side effects also imply increased patient tolerance and compliance, resulting in maximum therapeutic effect, without negatively impacting quality of life. Propolis, one of the most well-known honeybee products, has been used in folk medicine for its numerous health benefits since the dawn of civilization [2,3].

It is now used as a raw material for extracts that can be used as active pharmaceutical ingredients (APIs). Honeybees collect raw propolis, a natural glue-like substance, primarily from flower and leaf buds of various plant species. Beeswax and pollen are the most common components of propolis in general; it is also made up of plant resins and essential oils.

Polyphenols, terpenes, esters, amino acids, vitamins, minerals, and sugars have all been found in propolis [1,2,3]. Propolis is considered an animal-derived product because it is incorporated into the beehive by the honey bee’s digestive system, especially its saliva. Honey bees primarily use propolis as a thermal isolation material to seal the cracks in wood walls and other hive parts and strengthen the wax combs themselves.

Extracted propolis is capable of combating a wide range of microorganism types, including yeasts, fungi, viruses, bacteria, and even parasites, with the most significant effect on Gram-positive bacteria such as *Streptococcus* spp., *Staphylococcus aureus*, *Bacillus subtilis*, and *Enterococcus faecalis*, as well as *Candida* spp. [4,5].

This plant’s antimicrobial activity is frequently attributed to its polyphenols, which increase bacterial membrane permeability, disrupt membrane potential, reduce ATP production, and minimize bacterial mobility [3]. Some damage biofilms and have anti-quorum sensing activity [6].

The polyphenols, resins, flavonoids, essential oils, fatty acids, and wax found in propolis are just a few of the compounds that make up the honey bee’s best friend [1]. The use of propolis to treat various conditions in small animal species is beginning to play an essential role in the currently available treatments. Its use appears to be a cost-effective and effective treatment, with no side effects [7]. *Staphylococcus aureus* is an common bacteria that cause disease in dogs and is thought to be a reservoir for drug-susceptible and drug-resistant bacteria such as *S. aureus* [8,9]. This has led to the development of new alternative treatments, one of which is ethanolic extracts of propolis for topical applications, which have proven to be effective and inexpensive.

All *Staphylococcus* species that cause infections in humans and domestic animals can develop resistance to antibiotics [8,9,10,11]. The predisposition for staphylococci to develop antimicrobial resistance is a cause of major concern in both human and veterinary medicine [12,13].

## 2. Materials and Methods

### 2.1. Romanian Propolis

This study used standardized crude Romanian propolis samples provided by ApiLand SRL (Baia Mare, Romania) and their ethanolic extracts (ERPE), prepared in the medical botany laboratory of the Faculty of Veterinary Medicine in Timișoara, Romania. The components of propolis can identify in the plant sources visited by bees. In Romania, the primary sources of propolis include hazelnut (*Alnus* spp. and *Corylus* spp.), birch (*Betula* spp.), oak (*Quercus* spp.), poplar (*Populus* spp.), and willow (*Salix* spp.).

The crude dried propolis was ground to obtain a fine powder, from which samples of 2 g were taken, each mixed with 10 mL of 95% ethanol solution. The resulting mixture was kept at room temperature for seven days, with periodic strong handshaking. After extraction, the mixture was centrifuged at 26,000× *g* for 30 min and filtrated with filter paper (Whatman No. 4). A rotary evaporator was used at 50 °C with low pressure, to evaporate the remaining ethanol. The sample was kept at 4 °C in the dark until use. EERP was diluted in dimethyl sulfoxide (DMSO) for antimicrobial experiments within a final DMSO concentration lower than 1%, non-lethal for microorganisms [14]. The supernatant obtained is called ethanolic propolis extract. The alcohol level and concentration of dry extract in the EERP sample were 20% (*w*/*v*) [15].

The propolis sample was characterized using high-performance liquid chromatography (HPLC). The crude propolis and extracts were tested at concentrations ranging from 1% to 10% (diluent: Prosolv—Merck^®^ chromatographic grade methanol; Merck KGaA, Darmstadt, Germany). Analyses were carried out on an i-Series Integrated (u)HPLC System (Shimadzu, Columbia, SC, USA) automatic injector and high-efficiency liquid chromatography with a network of photodiodes.

Chromatographic analyses were performed according to the methods described by Bankova et al. [16].

Our study used chemicals (formic acid, methanol, and flavonoid standards) purchased from Fluka AG (Buch, Switzerland). P-coumaric acid, caffeic acid, rutin, quercetin, apigenin, chrysin, p-coumaric acid, ferulic acid, pinocembrin and phenolic compounds were supplied by Sigma Aldrich (Steinheim, Germany). Ultrapure water was used to prepare standard solutions and blanks (MicroPure water purification system, 0.055 µS/cm, TKA, Thermo Ficher Scientific, Niederlbert, Germany).

### 2.2. Staphylococcus aureus Clinical Strains

A total of 38 clinical *Staphylococcus aureus* strains used in our study were obtained from the Department of Infectious Diseases and Preventive Medicine, the Faculty of Veterinary Medicine Timisoara, the Bacterial Diseases diagnostic laboratory (B.6.a). *Staphylococcus aureus* strains were isolated from skin samples collected from dogs with superficial dermatitis between November 2018 and April 2020 in an urban area of Timisoara City, western Romania. The dogs that made up this study’s subject were dogs with owners who had requested veterinary services at the University Veterinary Clinics of the Timisoara Faculty of Veterinary Medicine.

During the dermatological evaluation, skin samples were harvested from dogs showing lesions at the skin level. All of the dogs included in this study exhibited one of the following lesions of the skin level: erythema, scaling, alopecia, pruritus, crusts, pustules, or hyperkeratosis. In addition, it is important to mention that no dogs included in the study received any treatment based on antibiotics.

The samples from dogs were collected in compliance with all biosecurity measures described in agreement with the regulations imposed by the Romanian Veterinary College (protocol numbers 34/1.12.2012), according to current practice at the University Veterinary Clinics the Faculty of Veterinary Medicine from Timisoara, and were taken with the consent of the pets’ owners and the approval of the Ethical Commission of the Banat’s the University of Agricultural Science and Veterinary Medicine from Timișoara, Romania [17].

*Staphylococcus aureus* was isolated using conventional methods (including Gram staining, colony morphology, hemolysis, test for catalase, coagulase activity, and anaerobic fermentation of mannitol) as mentioned in the protocols described by Lindsay et al. [10] and Fernandes-Queiroga-Moraes et al. [12], and confirmed with a molecular test. Catalase-positive and coagulase-positive staphylococcal isolates were identified using a Vitek Compact 2 system (bioMerieux, Marcy l’Etoile, France), according to the manufacturer’s instructions. All bacterial strains were stored in Trypticase Soy Broth (TSB) medium (ThermoFisher Scientific, Oxoid Ltd., Hampshire, UK) with 20% of glycerol at −86 °C, until further analyses were performed [18].

The Pure LinkTM Genomic Lysis/Binding Buffer (Thermo Fisher Scientific, Horsham, UK) boiling method with a freshly-prepared proteinase K solution (10 mg/mL) was used to confirm the identifications of *Staphylococcus aureus*, as previously described by Rantakokko-Jalava and Jalava [19].

The *sa-f* and *sa-r* genes of *Staphylococcus* 16S-1 were amplified using genus-specific primers: 5′-GTGCCAGCAGCCGCGGTAA-3′ and 5′-AGACCCGGGAACGTATTCAC-3′. To identify the species *Staphylococcus aureus*, the primers for the *nucA* nuclease gene employed in our work were *nuc-1*: 5′-TCAGCAAATGCATCACAAACAG-3′ and *nuc-2*: 5′-CGTAAATGCACTTGCTTCAGG-3′. In order to molecularly evidence the *mecA* gene, in our study, the primers *mecA-1*: 5′-GGGATCATAGCGTCATTATTC-3′ and *mecA-2*: 5′-AACGATTGTGACACGATAGCC-3′ were utilized, as previously described by Degi et al. [8].

The positive control strain was *Staphylococcus aureus* ATCC^®^ 23235^TM^.

All clinical *Staphylococcus aureus* strains were tested for antimicrobial susceptibility by broth microdilution method-based analysis, using Muller Hinton agar (ThermoFisher Scientific, Oxoid Ltd., Hampshire, UK) and a commercially available disk containing gentamicin (CN, 30 μg; range tested: 2–16 μg/mL) and enrofloxacin (ENR, 5 μg; range tested: 0.25–16 μg/mL) (ThermoFisher Scientific, Oxoid Ltd., Hampshire, UK), according to CLCI recommendations [20]. The testing used a bacterial suspension with 1.5 × 10^8^ CFU/mL.

### 2.3. Inocula

Inoculums were prepared by direct suspension of a specific colony, according to CLSI recommendations [21,22,23]. *Staphylococcus aureus* strains were inoculated on Mueller Hinton agar and incubated under aerobic conditions at 37 °C for 24 h. Bacterial suspensions in sterile saline solution were prepared directly by sampling and homogenizing the specific colony. The turbidity of these suspensions was adjusted to 0.5 McFarland (1.5 × 10^8^ CFU/mL).

We used 10^8^ CFU/mL to approximate the organism density commonly associated with various infections. Infection with *Staphylococcus aureus* frequently results in a high bacterial density (10^8^–10^10^ cells/g of tissue) [24].

### 2.4. Determination of Minimum Inhibitory Concentration

Our study determined that the lowest concentration of the assayed antimicrobial agent (minimal inhibitory concentration, MIC) was based on broth and agar dilution methods [25,26].

The minimal inhibitory concentrations (MICs) of Romanian propolis against the tested strains were determined using ethanolic propolis extracts at the following concentrations: 0, 0.0625, 0.125, 0.25, 0.5, 1, 2, 4, 6, 8, 10, 12, 14, and 16 µg/mL. Serial dilutions of ethanol extracted from Romanian propolis were prepared using purified sterile water (MicroPure water purification system, 0.055 µS/cm, TKA, Thermo Ficher Scientific, Niederlbert, Germany). Additionally, control plates containing serial concentrations of ethanolic alcohol solution were examined. The HMI-60&24 Multi-point Inoculator (Yima Optoelec Co., Ltd., Xi’an, Shaanxi, China) was used to inoculate the strains. Each test was conducted three times.

Agar dilution is a technique that entails the incorporation of various concentrations of an ethanolic extract of Romanian propolis into a Muller-Hinton agar medium. A standardized number of cells is then applied to the agar plate’s surface.

The antimicrobial agent stability and the method’s reproducibility were evaluated simultaneously in all the analyses performed, using an inoculation of the strains ATCC of *S. aureus*: *S. aureus* subsp. *aureus* Rosenbach—29213 (Methicilin sensible *Staphylococcus aureus*/MSSA) and *S. aureus* subsp. *aureus* Rosenbach—33511 (Methicilin resistant *Staphylococcus aureus*/MRSA), in the collection of the Research Laboratory in Bacterial diseases of animals, Faculty of Veterinary Medicine, Timișoara, Romania.

Immediately after homogenization of inoculum, using an automatic pipette (Eppendorf Research Plus—IVD, 100–1000 µL, Merck KGaA, Darmstadt, Germany) and disposable tips (Eppendorf^®^ epT.I.P.S. standard, Merck KGaA, Darmstadt, Germany), a volume of 350 µL was distributed in the respective wells of an inoculation plate of the HMI- 60&24 Multi-point Inoculator (Yima Optoelec Co., Ltd., Xi’an, Shaanxi, China). Each strain was inoculated only once during the experiment.

To determine the strain’s viability, inoculation was performed initially on plates without antibiotics (blanks), followed by inoculations onto plates containing increasing concentrations of diluted ethanolic propolis extract. Finally, a second growth-promoting plate was inoculated, to ensure that no contamination or significant antimicrobial agent load was introduced during the inoculations. The plates were inverted and incubated at 37 °C for 24 h under sterile conditions. The MIC is expressed in micrograms per milliliter (µg/mL) [3].

### 2.5. Determination of Minimum Bactericidal Concentration

MBC was determined by dipping sterile swabs into tubes containing propolis concentrations more significant than the MIC and inoculating them onto the agar medium. On the agar, there was no visible growth of MBC.

### 2.6. Statistical Analyses

All laboratory determinations were conducted three times in triplicate; data are presented as mean ± standard deviation (SD). To determine the percentage of the variation attributable to factors such as bacterial strains, time of incubation, and concentrations, the results concerning the bacterial growth were analyzed using a three-way analysis of variance (ANOVA test). All statistical analyses were performed using the OriginPro software package (version 2016, OriginLab Corporation, Northampton, MA, USA), assuming a significant level difference of *p* < 0.001.

## 3. Results

### 3.1. Ethanolic Extract of Romanian Propolis Analyses

Table 1 represents the result of a typical HPLC quantitative analysis of an ethanolic extract of Romanian propolis. Quantitative and qualitative analyses of selected parameters identified flavonoids, phenolic acids, cinnamic acid, kaempferol, rutin, caffeic acid, caffeic acid phenethyl ester, p-coumaric acid, quercetin, apigenin, ferulic acid, pinocembrin, and chrysin.

Romanian propolis ethanolic extract is rich in polyphenols, which are also considered to be the most active (Table 1). In addition, the ethanolic extract of propolis also includes other phenolic compounds (e.g., artepillin C), which are responsible for its characteristic fragrance. The amount of artecillin C (3.5-diprenyl-4-hydroxycinnamic acid) is the highest (mean 12.335 mg/g). The flavonoid group includes chrysin (mean 0.896 mg/g), pinocembrin (mean 0.876 mg/g), apigenin (mean 3.794 mg/g), kaempferol (mean 0.312 mg/g), quercetin (mean 1.538 mg/g), and quercitin hydrate (mean 1.621 mg/g). Another critical group of compounds of ethanolic extract of propolis are aromatic acids, among which the most common are ferulic (mean 0.325 mg/g), cinnamic (mean 3.439 mg/g), cafeic (mean 0.175 mg/g), and propenoic acids (mean 1.119 mg/g), and acrylic acid (mean 0.905 mg/g). In our study, the content of flavonoid compounds in Romanian propolis ranged from 0.312 to 3.794 mg/g.

This diversity of the chemical composition gives Romanian propolis an additional advantage as an antibacterial agent. The combination of many active ingredients and their presence in various proportions prevents bacterial resistance from occurring.

As previously stated, propolis has a highly complex chemical composition that varies according to the flora found in the areas from which it is collected. According to the chemical analysis, Romanian propolis was classified as a temperate zone propolis and correlated with the results obtained in other studies conducted in Romania [1,27,28,29].

The HPLC analysis showed that the flavonoid, phenolic acid, and other compounds of crude Romanian propolis and their ethanolic extract are similar to other ethanolic extracts of propolis, mainly of European origin [15,18,30,31,32,33].

We concluded that Romanian propolis possesses significant amounts of biologically active compounds, especially flavonoids and phenolic acids, and can be subjected to other validated methods for European propolis, including Romanian propolis. However, the interactions among the different chemical compounds in the propolis sample are essential when considering its antibacterial effects against pathogens, such as *Staphylococcus aureus*.

### 3.2. Bacteriological Study

According to the standard and molecular method, 38 clinical strains belonged to the *Staphylococcus aureus* species. Catalase, coagulase-positive, and mannitol fermenting staphylococcus strains were identified biochemically, using the Vitek Compact 2 System. The molecular species identification of staphylococci by PCR technique, using specific primers, *nuc-1* and *nuc-2*, respectively, confirmed that all clinical isolates were *Staphylococcus aureus* strains, of which 12 (31.57%, 12/38) showed positive results for the presence of the *mecA* gene.

### 3.3. Determinations MIC and MBC Values

The MBC values of propolis were evaluated by sub-culturing about 5–10 µL in wells with a concentration equal to or higher than MIC on a blood agar plate for bacteria. This effect was compared to synthetic antibiotics’ inhibition of bacterial growth, gentamicin (MIC GM 30 µg; 4–16 µg/mL; ThermoFisher Scientific, Oxoid Ltd., Hampshire, UK) and enrofloxacin (MIC ENR 15 µg; 0.5–4 µg/mL; ThermoFisher Scientific, Oxoid Ltd., Hampshire, UK) were used as positive controls. These antibiotics are frequently used in veterinary dermatology.

The susceptibility results of the clinical and reference *Staphylococcus aureus* strains to EERP (MIC and MBC values in µg/mL) are shown in Table 2, Table 3 and Table 4.

The MICs of the EERP against 38 clinical *Staphylococcus aureus* strains were determined using the agar microdilution method.

After 24 h of incubation, the analysis of bacterial growth showed that all MRSA and MSSA strains were inhibited at concentrations (MIC) ranging from 6 to 10 µg/mL (Mean + STDEV.P: 9.483 µg/mL). For MRSA strains, mean + STDEV.P MIC values were 9.632 µg/mL, and for MSSA strains, 9.467. The MBC mean + STDEV.P value was for MRSA and MSSA clinical strains and found to be 12.396 and 12.417, respectively. 

The three-way ANOVA indicated that the growth of all clinical *Staphylococcus aureus* strains was significantly affected by EERP concentration (*p* < 0.001) and incubation time (*p* < 0.001). The interaction between these factors was also significant (*p* < 0.001).

## 4. Discussion

This present study determined that an ethanolic extract of Romanian propolis samples had highly biological effects in vitro against clinical *Staphylococcus aureus* strains. Since increasing resistance to antibiotics may lead to the failure of the therapy of dermatological diseases in dogs, we investigated the antibacterial activity of a natural Romanian propolis product. All tested clinical *Staphylococcus aureus* strains and *Staphylococcus aureus* reference strains (ATCC) were determined to be susceptible to this bee product, and the MIC values ranged from 6 to 10 µg/mL, regardless of the ethanolic extract of Romanian propolis.

Antimicrobial actions have been frequently recognized among the various biological activities of propolis extracts. Propolis ethanolic extracts have been reported to be efficient against a broad spectrum of bacteria, particularly Gram-positive bacteria [34,35]. The antibacterial action of Romanian propolis has been linked to the presence of phenolics, flavonoids, and caffeic acid derivatives [25,27,36,37]. Ethanol extraction is the most common method for producing propolis extracts. This approach can be used to make low-wax propolis extracts with a high concentration of physiologically active chemicals. Artepillin C (3,5-diphenyl-p-coumaric acid) is one of the phenolic compounds naturally found in propolis (prenyl derivative of p-coumaric acid) [3]. Additionally, these extracts demonstrated significant antibacterial activity against MRSA *S. aureus* [38]. Kaempferide also exhibits antimicrobial activity against microbes associated with skin infections, such as *S. aureus* [36]. Propolis ethanolic extracts with high concentrations of kaempferide, artepillin-C, drupanin, and p-coumaric acid demonstrated antioxidant and antibacterial activity against *S. aureus*, *S. saprophyticus*, *Listeria monocytogenes*, and *E. faecalis* [39].

Ethanolic extract of propolis (EEP) is most commonly used in antimicrobial activity analyses [3]. It is preferable to compare the effect of EEP on *S. aureus* as a representative of Gram-positive bacteria, based on geographical origin [3].

Many previous studies have looked at the antimicrobial properties of propolis from temperate regions of Europe [1,3,14,40,41]. However, few studies have linked this effect to the specific chemical composition or the polyphenol content [27]. The results show that ethanolic extracts of Romanian propolis had an antibacterial effect on Gram-positive bacteria. European propolis extract had the most potent antibacterial effect on *Staphylococcus aureus*. There are a lot of different bioactive compounds in European propolis ethanolic extracts, including the Romanian propolis, and it is thought that these compounds work together to make the extract antibacterial. This could be a good thing, because different kinds of microorganisms cannot become resistant to many different kinds of biological compounds [1].

The antibacterial properties of propolis have a long and distinguished history in science. According to a recent review by Przybyek and Karpinski [3], a total of 600 aerobic and anaerobic bacterial strains have been studied to determine the effect of propolis. The review in [3] provides information on which bacterial species are susceptible to propolis action and the minimum inhibitory concentration values (a minimum concentration at which no microorganism growth can be observed in the assays). The antimicrobial activity against Gram-positive bacteria of propolis has been demonstrated in numerous studies to be stronger than that of other bacteria [3,41,42]. Gram-negative bacteria’s outer membranes contain bacterial hydrolytic enzymes, which may compromise and reduce the efficacy of propolis’ active ingredients [43,44]. Propolis’ phenolic and flavonoid content is commonly linked to its antimicrobial properties. However, the concentration of these components does not always correlate with observed antimicrobial activity in vitro, as demonstrated by Bridi et al. [45]. Propolis’ ability to prevent bacterial resistance can be attributed to the wide variety of active ingredients in various combinations and concentrations [46]. The composition of propolis and its antibacterial properties appear to be influenced by its geographic origin [5].

The most popular honeybee species is the European honey bee, *Apis mellifera*. It has been shown that the variety of bee affects the antibacterial activity of propolis collected from the same apiary.

The biological activities of propolis are attributed to a variety of major chemical constituents, including phenolic acids, phenolic acid esters, flavonoids, terpenoids, artepillin C, caffeic acid, chrysin, galangin quercetin, apigenin, kaempferol, pinobanksin 5-methyl ether, pinobanksin, pinocembrin, and pinobanksin 3-acetate [20]. High-performance liquid chromatography (HPLC) was, and still is, the preferred separation technique for analyzing natural products [16,46]. HPLC separation is mainly dependent on the different affinities between the propolis compounds and the stationary phase. For a particular application, the chemical properties of the packing and physical properties of the column (e.g., particle size and column dimensions) need to be taken into account.

Reversed-phase HPLC is doubtlessly the most widely used chromatographic method in propolis analysis [47,48,49].

Due to the matrix’s complexity, a disadvantage of using these columns is the lengthy runs required; frequently exceeding 50 min per run. The chromatographic conditions of the HPLC methods include, almost exclusively, the use of UV-visible diode-array detection (DAD) using the device Infinity II Diode Array Detector WR-1260 (Agilent, Santa-Clara, CA, USA) with spectral data for all peaks acquired in the range of 150–950 nm. However, 280 nm is the most generic wavelength for phenolic compounds, due to the high molar absorptivity of the different phenolic classes at that wavelength [16,48].

The eluent comprises a binary solvent system containing acidified water (solvent A) combined with a polar organic solvent (solvent B). Gradient elution is usually mandatory in recognition of the complexity of the propolis chemical profile. A quantity of 0.1% formic or acetic acid can be added to water (as solvent A), and acetonitrile or methanol (as solvent B) are commonly used in propolis analysis. While 0.1% formic acid is the most suitable when using an MS detector.

We recommend that bee varieties and subspecies be considered together with geographical factors and plant species around the beehive in future studies on propolis. Characterization of propolis from various locations and plant sources is warranted, to define acceptable quantitative standards for different types of propolis. Furthermore, the biological activities of each type of propolis need to be correlated with their chemical composition, and eventually, standardized products should be used in clinical studies.

According to the World Health Organization, between 70% and 95% of underdeveloped nations use natural products as a therapeutic option [50]. The composition of propolis is exceedingly complicated and changeable, containing 50–55% resin, 30% beeswax, 10–15% essential oils, and 5% pollen [51,52]. Physical qualities of propolis, such as the color, aroma, and consistency, are influenced by factors such as geographic origin, types of vegetal sources, collecting period, and season [53]. Propolis’ chemical makeup is highly dependent on its geographical location. The botanical origins and chemical makeup of propolis have a close relationship, resulting in a wide range of propolis ingredients [53].

EEP from Turkey, Taiwan, and Oman exhibited the highest activity against *S. aureus*, with MIC values of 8, 10, and 81 µg/mL, respectively. The lowest levels of activity were observed in propolis samples from Chile, Australia, and Germany; and EEP, in this case, had a MIC of 1445, 1200, and 750 µg/mL, respectively. Even though it has conducted the most research on propolis, Brazil was ranked in the middle for both Gram-positive and Gram-negative bacteria [3].

Our study on the influence of EERP on MSSA and MRSA clinical strains showed no significant differences, both the MIC and MBC values obtained for these strains were relatively similar. This may suggest that EERP demonstrates an essential anti-staphylococcal activity not associated with β lactam antibiotics.

The results obtained in the in vitro study encouraged us to perform an in vivo test.

## 5. Conclusions

The ethanolic extracts from Romanian propolis were active in vitro against clinical *Staphylococcus aureus* strains. They are considered viable synthetic products for treating superficial dermatitis in dogs produced by staphylococci.

The emergence of resistant *S. aureus* strains in dogs has prompted the development of new therapeutic agents, such as propolis, which is regarded as an effective natural product with a diverse set of biological properties, including antimicrobial activity. The ERPE sample contained the highest concentration of phenols and flavonoids, resulting in antibacterial sensitivity of all clinical *S. aureus* strains tested. Romanian propolis’s biological potential may make it an ideal candidate for developing new therapeutic, cost-effective, and side-effect-free antimicrobial agents for infectious dermatitis in dogs.

## Figures and Tables

**Table 1 vetsci-09-00299-t001:** The chemical composition of a crude sample of Romanian propolis and its ethanolic extract was determined (quantitative analyses).

Chemical Composition	Crude Romanian Propolis Sample		Ethanolic Romanian Propolis Extract Samples (ERPE)	
	Mean (mg/g)	Standard Deviation	Mean (mg/g)	Standard Deviation
phenyl acrylic acid 3-[4-hydroxy-3-(oxo butyl)]	3.465	0.049	0.905	0.012
phenyl cinnamic acid-5-3-prenyl-3(E)-(4-hydroxy-3-methyl-2-butenol)	0.129	0.007	0.682	0.002
cinnamic acid—3-prenyl-4-(2-methylpropionyloxi)	0.789	0.023	0.187	0.014
3-prenyl-4-dihydrocynamoiloxicinnamic acid	0.224	0.003	0.202	0.000
Ferulic acid	0.318	0.004	0.325	0.016
3-prenyl-4-hydroxycinnamic acid	1.991	0.011	3.439	0.088
2.2-dimethyl-6-carboxyethenyl-2H-1-benzopirane	3.993	0.001	2.782	0.046
2.2-dimethyl-8-prenyl-2H-1-benzopirano-6-propenoic acid	1.569	0.032	1.119	0.001
(E)-3-{4-hydroxy-3-[(E)-4-(2.3)-dihydrocynamoiloxi-3-methyl-2-butenyl]-5-prenylphenyl-2-propenoic acid (artepillin C)	1.910	0.014	0.658	0.432
Chrysin (5,7-Dihydroxyflavone)	1.398	0.028	0.896	0.214
3.5-diprenyl-4-hydroxycinnamic acid	21.524	0.392	12.335	0.056
Apigenin (5,7-Dihydroxy-2-(4-hydroxyphenyl)-4-benzopyrone)	9.134	0.198	3.794	0.008
p-coumaric acid	11.321	0.819	6.183	0.072
Quercetin (2-(3,4-dihydroxyphenyl)-3,5,7-trihydroxy-4H-chromen-4-one)	4.861	0.043	1.538	0.033
Kaempferol	0.418	0.028	0.312	0.001
Quercetin-3-rutinoside hydrate (Rutin hydrate)	4.685	0.032	1.621	0.415
Caffeic acid	0.342	0.002	0.175	0.004
1—caffeoylquinic acid	0.495	0.485	0.314	0.005
2—caffeoylquinic acid	0.725	0.594	0.062	0.028
3—caffeoylquinic acid	1.427	0.484	0.612	0.003
4—caffeoylquinic acid	2.619	0.001	0.117	0.000
Pinocembrin	0.536	0.001	0.876	0.067
Total	69.897	-	32.811	-

**Table 2 vetsci-09-00299-t002:** MIC and MBC values of ethanolic extract of Romanian propolis and two antibacterial drugs (µg/mL).

MRSA/MSSA	Ethanolic Extract of Romanian Propolis		Gentamicin (30 µg)		Enrofloxacin (15 µg)	
	MIC (µg/mL)	MBC (µg/mL)	MIC (Sensible ≤ 4; Resistant ≥ 16 µg/mL *)		MIC (Sensible ≤ 0.5; Resistant ≥ 4 µg/mL *)	
			Sensible	Resistant	Sensible	Resistant
*Staphylococcus aureus* 1—MSSA	8	12	2	-	0.225	-
*Staphylococcus aureus* 2—MRSA	6	10	-	24	-	6
*Staphylococcus aureus* 3—MSSA	8	10	-	18	0.125	-
*Staphylococcus aureus* 4—MSSA	10	14	2	-	0.75	-
*Staphylococcus aureus* 5—MRSA	10	12	-	18	0.5	-
*Staphylococcus aureus* 6—MRSA	8	10	-	22	0.125	-
*Staphylococcus aureus* 7—MSSA	6	8	1	-	0.75	-
*Staphylococcus aureus* 8—MSSA	10	12	2	-	0.225	-
*Staphylococcus aureus* 9—MSSA	6	8	4	-	-	4
*Staphylococcus aureus* 10—MRSA	6	8	-	24	0.5	-
*Staphylococcus aureus* 11—MSSA	8	12	2	-	0.125	-
*Staphylococcus aureus* 12—MSSA	8	10	4	-	0.75	-
*Staphylococcus aureus* 13—MSSA	10	14	1	-	0.125	-
*Staphylococcus aureus* 14—MSSA	6	8	4	-	0.225	-
*Staphylococcus aureus* 15—MSSA	6	8	-	16	0.25	-
*Staphylococcus aureus* 16—MSSA	8	10	2	-	0.25	-
*Staphylococcus aureus* 17—MRSA	10	14	2	-	-	6
*Staphylococcus aureus* 18—MSSA	8	10	4	-	0.125	-
*Staphylococcus aureus* 19—MSSA	8	10	-	18	0.225	-
*Staphylococcus aureus* 21—MRSA	6	8	-	24	-	8
*Staphylococcus aureus* 22—MSSA	6	6	2	-	0.5	-
*Staphylococcus aureus* 23—MSSA	8	10	2	-	0.125	-
*Staphylococcus aureus* 24—MRSA	10	14	4	-	0.25	-
*Staphylococcus aureus* 25—MRSA	10	12	4	-	-	6
*Staphylococcus aureus* 26—MSSA	8	10	1	-	0.125	-
*Staphylococcus aureus* 27—MSSA	6	8	2	-	0.225	-
*Staphylococcus aureus* 28—MSSA	6	6	-	18	0.5	-
*Staphylococcus aureus* 29—MSSA	10	12	1	-	0.25	-
*Staphylococcus aureus* 30—MRSA	8	8	4	-	-	8
*Staphylococcus aureus* 31—MSSA	10	12	2	-	-	6
*Staphylococcus aureus* 32—MRSA	6	8	4	-	0.125	-
*Staphylococcus aureus* 33—MSSA	6	8	-	22	0.25	-
*Staphylococcus aureus* 34—MSSA	8	10	1	-	0.5	-
*Staphylococcus aureus* 35—MRSA	8	8	2	-	0.125	-
*Staphylococcus aureus* 36—MSSA	10	12	-	16	0.125	-
*Staphylococcus aureus* 37—MSSA	10	14	2	-	0.25	-
*Staphylococcus aureus* 38—MRSA	8	10	4	-	0.225	-
*Staphylococcus aureus* ATCC 25923—MSSA	8	10	1	-	0.225	-
*Staphylococcus aureus* ATCC 43300—MRSA	8	10	2	-	0.225	-
Total (Mean + STDEV.P)	9.483	12.344	3.339	16	0.5	8

* CLSI: Performance standard for antimicrobial disk susceptibility test, 13th ed. CLSI M02 Wayne, PA, Clinical and Laboratory Standard Institute, 2018.

**Table 3 vetsci-09-00299-t003:** *Staphylococcus aureus* MRSA strains are susceptible to EERP (MIC and MBC and Mean + STDEV.P values in µg/mL).

Microorganism (No. of Strain)	MIC (µg/mL)	MBC (µg/mL)
*Staphylococcus aureus* 2—MRSA	6	10
*Staphylococcus aureus* 5—MRSA	10	12
*Staphylococcus aureus* 6—MRSA	8	10
*Staphylococcus aureus* 10—MRSA	6	8
*Staphylococcus aureus* 17—MRSA	10	14
*Staphylococcus aureus* 21—MRSA	6	8
*Staphylococcus aureus* 24—MRSA	10	14
*Staphylococcus aureus* 25—MRSA	10	12
*Staphylococcus aureus* 30—MRSA	8	8
*Staphylococcus aureus* 32—MRSA	6	8
*Staphylococcus aureus* 35—MRSA	8	8
*Staphylococcus aureus* 38—MRSA	8	10
Total (Mean + STDEV.P)	9.632	12.396

**Table 4 vetsci-09-00299-t004:** *Staphylococcus aureus* MSSA strains are susceptible to EERP (MIC and MBC and Mean + STDEV.P values in µg/mL).

Microorganism (No. of Strain)	MIC (µg/mL)	MBC (µg/mL)
*Staphylococcus aureus* 1—MSSA	8	12
*Staphylococcus aureus* 3—MSSA	8	10
*Staphylococcus aureus* 4—MSSA	10	14
*Staphylococcus aureus* 7—MSSA	6	8
*Staphylococcus aureus* 8—MSSA	10	12
*Staphylococcus aureus* 9—MSSA	6	8
*Staphylococcus aureus* 11—MSSA	8	12
*Staphylococcus aureus* 12—MSSA	8	10
*Staphylococcus aureus* 13—MSSA	10	14
*Staphylococcus aureus* 14—MSSA	6	8
*Staphylococcus aureus* 15—MSSA	6	8
*Staphylococcus aureus* 16—MSSA	8	10
*Staphylococcus aureus* 18—MSSA	8	10
*Staphylococcus aureus* 19—MSSA	8	10
*Staphylococcus aureus* 22—MSSA	6	6
*Staphylococcus aureus* 23—MSSA	8	10
*Staphylococcus aureus* 26—MSSA	8	10
*Staphylococcus aureus* 27—MSSA	6	8
*Staphylococcus aureus* 28—MSSA	6	6
*Staphylococcus aureus* 29—MSSA	10	12
*Staphylococcus aureus* 31—MSSA	10	12
*Staphylococcus aureus* 33—MSSA	6	8
*Staphylococcus aureus* 34—MSSA	8	10
*Staphylococcus aureus* 36—MSSA	10	12
*Staphylococcus aureus* 37—MSSA	10	14
Total (Mean + STDEV.P)	9.467	12.417

## Data Availability

Not applicable.

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
