# Peer review of "Antibacterial Activity of Romanian Propolis against Staphylococcus aureus Isolated from Dogs with Superficial Pyoderma: In Vitro Test"

_vetsci, 2022, doi:10.3390/vetsci9060299_

Round 1

Reviewer 1 Report

In this manuscript “Antibacterial activity of Romanian propolis against S. aureus…” , Degi et al describe extracting biologically active components of propolis for testing against S. aureus isolates from dogs suffering various dermatologic manifestations of infection.  This paper contributes to the body of work on the biological activities of propolis, while not novel, this work adds to the knowledge base of propolis' antimicrobial activities. 

Major Concerns:

Not clear from manuscript whether authors or investigators got permission from dog owners for experimental use of these samples. This must be clarified before publication.

Saying that natural products are a new source of therapies on Page 1 Lines 32-39 is untrue and in direct contradiction to what you say in Lines 40-41. Please clarify.

Page 1 Lines 32-44 are missing citations.

Page 2 Line 64 needs to be clarified. Not all staph are drug resistant but all species have picked up drug resistance at one time or another in one place or another

Minor concerns:

HPLC is misspelled as HPCL throughout manuscript (results and discussion sections especially)

It is not appropriate to have a single sentence be its own paragraph in intro or in methods section. Please check and refine structure throughout manuscript.

Suggest thorough review of grammar. For example on Page 3 Lines 117-120, the writing is cumbersome and not easy to follow for the readers. 

Author Response

Dear Reviewer,

The manuscript has been revised for better readability according to the suggestions and recommendations.

The latest corrections in the revised manuscript text are marked in blue to be more easily identified and traced.

With due respect for your hard work and expertise,

                                                Sincerely with friendship, Lecturer Dr. Dégi János

Reviewer 2 Report

1. Overall, English phrases and words are sometimes not understandable. Followings are examples. 1) Abstract, line 12 : "...is associated...".  2) line 140, "emphasize", 3) line 143, "highlight", etc. Please chech wording throughout the manuscript and correct. 

2. Unnecessary duplication should be deleted.  In section 3.2, contains duplication of those in Methods section. This section title, "molecular study" is strange.  This is just detection of nuc and mecA by PCR. 

3. result section 3.1 and Table 1 are not well explained in text. Among substances identified, 3,5-diprenyl-4-hydroxycinnamic acid is the most abundant. But its significance and comparison with other reported propolis are not discussed in Discussion. line 310-317, propolis in Europe was written, but any difference compared with the present "Romanian propolis" is not shown. These points should be added. In addition, unnecessary descriptions should be deleted.

4. Authors measured MICs/MBCs of Romanian propolis to MRSA/MSSA. In Discussion section, chemical activity of European propolis extracts is shown as inhibition zone diameter. How did author compare the difference of antibacterial activity of Romanian propolis and European propolis? How did author evaluate efficacy of Romanian propolis? Such important points are missing.

5. Generally, this manuscript is unnecessarily long. Particularly, Materials and Methods section  is too long, thus should be shortened.  

Author Response

(The authors gave the same response as above.)

Round 2

Reviewer 1 Report

Thank you for addressing the concerns. I recognize the current form.

Author Response

Again, we wish to show our gratitude for the valuable time spent with our manuscript, improving its quality, and for all the priceless recommendations posed. We could also see and appreciate your high understanding of this topic and tried to take it all into account.

Best regards, Dr Degi Janos

Reviewer 2 Report

The revised version seems to be well written and become more understandable by readers. However, still there are phrases that should be grammatically corrected. I hope such minor corrections will be done by the publisher.

Only one comment to the authors is that line 134-135 is duplicated in line 237-239, in Results. The latter, Results portion must be deleted.  

Author Response

(The authors gave the same response as above.)
